# Upper Tract Urothelial Cancer: Guideline of Guidelines

**DOI:** 10.3390/cancers16061115

**Published:** 2024-03-11

**Authors:** Savio Domenico Pandolfo, Simone Cilio, Achille Aveta, Zhenjie Wu, Clara Cerrato, Luigi Napolitano, Francesco Lasorsa, Giuseppe Lucarelli, Paolo Verze, Salvatore Siracusano, Carmelo Quattrone, Matteo Ferro, Eugenio Bologna, Riccardo Campi, Francesco Del Giudice, Riccardo Bertolo, Daniele Amparore, Sara Palumbo, Celeste Manfredi, Riccardo Autorino

**Affiliations:** 1Department of Urology, University of L’Aquila, 67100 L’Aquila, Italy; pandolfosavio@gmail.com (S.D.P.); salvatore.siracusano@univaq.it (S.S.); 2Department of Neurosciences and Reproductive Sciences and Odontostomatology, University of Naples “Federico II”, 80131 Naples, Italy; simocilio.av@gmail.com (S.C.); achille-aveta@hotmail.it (A.A.); luiginap89@gmail.com (L.N.); 3Department of Urology, Changhai Hospital, Naval Medical University, Shanghai 200433, China; 4Urology Unit, University Hospital of Southampton NHS Trust, Southampton S016 6YD, UK; clara.cerrato01@gmail.com; 5Department of Precision and Regenerative Medicine and Ionian Area-Urology, University of Bari “Aldo Moro”, 70121 Bari, Italy; francesco-lasorsa96@libero.it (F.L.); giuseppe.lucarelli@inwind.it (G.L.); 6Department of Medicine and Surgery, Scuola Medica Salernitana, University of Salerno, 84084 Fisciano, Italy; pverze@gmail.com; 7Unit of Urology, Department of Woman, Child and General and Specialized Surgery, University of Campania “Luigi Vanvitelli”, 80138 Naples, Italy; carmeloquattrone@hotmail.it (C.Q.); manfredi.celeste@gmail.com (C.M.); 8Division of Urology, IRCCS—European Institute of Oncology, 20141 Milan, Italy; drmatteoferro@gmail.com; 9Department of Maternal-Child and Urological Sciences, Sapienza University Rome, Policlinico Umberto I Hospital, 00161 Rome, Italy; eugenio.bologna@uniroma1.it (E.B.);; 10Urological Robotic Surgery and Renal Transplantation Unit, Careggi Hospital, University of Florence, 50121 Firenze, Italy; riccardo.campi@gmail.com; 11Department of Urology, University of Verona, 37126 Verona, Italy; riccardobertolo@hotmail.it; 12Division of Urology, Department of Oncology, School of Medicine, University of Turin, San Luigi Hospital, Orbassano, 10043 Turin, Italy; danieleamparore@hotmail.it; 13Department of Molecular Medicine and Medical Biotechnologies, 80131 Naples, Italy; sarapalumbo.mail@gmail.com; 14Department of Urology, Rush University Medical Center, Chicago, IL 60612, USA

**Keywords:** upper tract urothelial carcinoma (UTUC), guidelines, diagnosis, management, follow-up

## Abstract

**Simple Summary:**

The creation of a guideline of guidelines could help physicians apprehend different approaches to malignancies and better-tailoring patients’ management. In this context, international guidelines on upper tract urothelial carcinoma (UTUC) have previously never been compared in their similarities and differences. The present effort could guide further researchers to provide more high-quality evidence to homogeneously assess, treat, and follow up patients affected by UTUC worldwide. Moreover, by highlighting similarities and differences, we aim to encourage more conscious and critical use of guidelines and push research to fill knowledge gaps.

**Abstract:**

Background: Upper tract urothelial carcinoma (UTUC) is a rare disease with a potentially dismal prognosis. We systematically compared international guidelines on UTUC to analyze similitudes and differences among them. Methods: We conducted a search on MEDLINE/PubMed for guidelines related to UTUC from 2010 to the present. In addition, we manually explored the websites of urological and oncological societies and journals to identify pertinent guidelines. We also assessed recommendations from the International Bladder Cancer Network, the Canadian Urological Association, the European Society for Medical Oncology, and the International Consultation on Bladder Cancer, considering their expertise and experience in the field. Results: Among all the sources, only the American Urologist Association (AUA), European Association of Urology (EAU), and the National Comprehensive Cancer Network (NCCN) guidelines specifically report data on diagnosis, treatment, and follow-up of UTUC. Current analysis reveals several differences between all three sources on diagnostic work-up, patient management, and follow-up. Among all, AUA and EAU guidelines show more detailed indications. Conclusions: Despite the growing incidence of UTUC, only AUA, EAU, and NCCN guidelines deal with this cancer. Our research depicted high variability in reporting recommendations and opinions. In this regard, we encourage further higher-quality research to gain evidence creating higher grade consensus between guidelines.

## 1. Introduction

Upper tract urothelial carcinoma (UTUC) accounts for 5–10% of urothelial cancers, of which bladder cancer (BC) represents by far the most frequent one [1]. Environmental factors such as smoking exposure and aristolochic acid are the most established risk factors [2]. The most common presentation symptoms are hematuria in almost 80% of cases and flank pain in 20%. In more advanced stages, constitutional symptoms—indicative of metastatic disease—often emerge. These include weight loss, fever, night sweats, anorexia, and hemoptysis [3,4,5]. However, UTUC is occasionally diagnosed incidentally during routine cross-sectional imaging with evidence of a collecting system mass [6].

The pauci-symptomatic nature of the disease is the main factor contributing to late detection and high rates of invasive cancer at diagnosis [7]. Indeed, only 15–20% of patients with urothelial BC present an invasive disease at diagnosis, whilst literature data show rates of approximately 70% in patients with UTUC [8]. Moreover, despite several similarities, UTUC has a more aggressive course than BC, with more than doubled rates of 5-year mortality (more than 50% vs. less than 25%, respectively) [9].

Due to specific challenges in disease identification and management, there is some disagreement regarding the management of this complex disease. This is also reflected by differences in the multiple guidelines from different scientific organizations that are available to the treating physician. In this study, we sought to systematically compare current international guidelines and recommendations and to identify both discrepancies and similarities regarding the management of UTUC.

## 2. Materials and Methods

### 2.1. Data Source

We conducted a search on MEDLINE/PubMed, Scopus, and Web of Science databases for guidelines on UTUC from 2010 to the present. In addition, we manually explored the websites of the main international urological and oncological societies to identify pertinent guidelines.

We explored the existence of guidelines on the topic by the American Urological Association (AUA), the American Society of Clinical Oncology (ASCO), the Society of Urologic Oncology (SUO), the European Association of Urology (EAU), the European Society for Medical Oncology (ESMO), the National Comprehensive Cancer Network (NCCN), and the National Institute for Health and Care Excellence (NICE). We also evaluated any recommendations on the topic from the International Bladder Cancer Network (IBCN), the International Bladder Cancer Group (IBCG), the Canadian Urological Association (CUA), and the International Consultation on Urologic Diseases (ICUD).

When multiple guidelines from the same scientific society were found, only the most recent ones were considered.

### 2.2. Guideline Evaluation

The assessment of guidelines was carried out using the Appraisal of Guidelines, Research, and Evaluation II (AGREE II) instrument, which offers a structured framework for comparing the quality of various guidelines [10,11]. This tool was employed by four authors independently (SDP (Assistant Professor); SC (Senior Urology Resident); AA (Urologist); CM (Researcher, PhD)). It is crucial to emphasize that all assessors involved in the evaluation process have been engaged in UTUC as professionals for at least 5 years, enhancing the reliability of their contributions to this research. The AGREE II instrument consists of 23 items categorized into six domains: scope and purpose, stakeholder involvement, rigor of development, clarity of presentation, applicability, and editorial independence (Appendix A). Assessors assigned ratings to each item on a scale of 1 to 7, where 1 represents ‘strongly disagree’ and 7 signifies ‘strongly agree’.

Following the AGREE II methodology, mean domain scores were determined by summing the scores for individual items within a domain and then scaling the total as a percentage of the maximum attainable score for that domain, considering the assessments of all four reviewers. These six domain scores remain separate and are not combined into a single overall quality score. Instead, appraisers assign a distinct overall score ranging from 1 to 7 and indicate whether they would recommend the guideline for use. An average overall score for each guideline was calculated based on the overall scores provided by the four reviewers.

### 2.3. Data Synthesis

The data were reported as in the original guidelines. Sums, means, percentages, and ranges were used to give a better overview of the results. No statistical tests were applied.

## 3. Results

### 3.1. International Guidelines on UTUC

Recent international guidelines on the topic were only available from the EAU, AUA, and NCCN.

#### 3.1.1. EAU Guidelines

UTUC clinical guidelines have been compiled by the non-muscle-invasive BC (NMIBC) guidelines panel to provide clinicians with evidence-based information and recommendations. The panel consists of 20 international multidisciplinary clinicians, including urologists, uro-oncologists, one radiologist, one pathologist, and, in the course of 2021, two patients joint representing the lay public.

The 2023 guidelines edition represents a limited update of the 2022 version [8]. The inaugural EAU guidelines for UTUC were released in 2011. The latest guidelines, along with a summary of any updates, are available on the following webpage: https://uroweb.org/guidelines/upper-urinary-tract-urothelial-cell-carcinoma/chapter/introduction (accessed on 1 January 2024).

Instead of using the modified Oxford Centre for Evidence-Based Medicine Levels of Evidence (LE), the system for evaluating the strength of the evidence (SE) now categorizes recommendations as either ‘weak’ or ‘strong’.

#### 3.1.2. AUA/SUO Guidelines

In 2021, the UTUC panel was established by the American Urological Association Education and Research, Inc. (AUAER) to work with the Society of Urologic Oncology (SUO) in creating clinical guidelines for treating localized or locally advanced UTUC. The latest update was released in April 2023 [9].

The AUA categorizes evidence strength into Grades A, B, and C. Grade A represents highly reliable evidence from well-conducted trials or strong observational studies. Grade B indicates moderately strong evidence with some weaknesses, while Grade C suggests evidence with serious deficiencies, small sample sizes, or inconsistency. In the absence of evidence, clinical principles and expert opinions (EOs) could be found as additional information.

#### 3.1.3. NCCN Guidelines

UTUC practical guidelines have been published on an annual basis as a chapter of BC Guidelines to complete the educational path of all healthcare providers, from nurses and pharmacists to urologists and oncologists.

Thanks to the clinical practice nature, the NCCN guidelines on BC briefly summarize treatment options on UTUC to guide healthcare providers in each step of diagnosis, treatment, and follow-up; on the other hand, they miss specific sections regarding epidemiology and risk factors.

The latest version of the guidelines was published in 2023 (third version) as an update of the 2022 version [12]. Recommendations were graded with the NCCN categories of evidence and consensus. These categories of evidence and consensus are ranged as follows: 1 (uniform consensus based on high-level evidence), 2A (uniform consensus based on lower-level evidence), 2B (consensus based on lower-level evidence), and 3 (major disagreement). Recommendations without high-level evidence were mainly based on expert opinion.

### 3.2. Assessment of Guidelines (AGREE II)

The authors separately evaluated guidelines with a measured overall reliability of 0.96 among them, indicating perfect agreement.

The mean scores of the various domains of the AGREE II were 87.3 (scope and purpose), 79.5 (stakeholder involvement), 95.2 (rigor of development), 96.4 (clarity of presentation), 98.5 (applicability), and 68.7 (editorial independence).

### 3.3. Epidemiology and Risk Factor

Both the EAU and AUA guidelines report UTUC as less common compared to BC, constituting about 5–10% of all urothelial carcinomas [13]. UTUC most commonly occurs in individuals aged between 70 and 90, with men being twice as susceptible as women [14].

The observed increase in incidence over recent years can be attributed to advancements in detection methods and enhanced survival rates for BC [15,16]. With regards to UTUC occurring after BC, a significant incidence rate has been reported by EAU, with a 7.5% incidence of UTUC in a 402-patient multicenter study of NMIBC cases treated with BCG over 50 months of follow-up. Intravesical recurrence and non-papillary tumors at TURB were identified as key predictors [17]. Even following radical cystectomy for muscle-invasive BC, there is an observed association rate, with 3–5% of patients developing metachronous UTUC [18]. Thus, there are shared risk factors and molecular pathways between UTUC and BC.

Guidelines acknowledge that environmental factors may contribute to the development of UTUC [19,20]. Overall, the EAU guidelines do not offer strong supporting evidence for this assertion, except in the case of aristolochic acid and smoking (LE: 2A, SE: weak), where tobacco exposure elevates the relative risk of UTUC from 2.5 to 7.0 [21].

Conversely, AUA/SUO offers a set of somewhat generic risk indications that could benefit from revision to provide a more detailed and in-depth description. This would offer more precise and comprehensive guidance, particularly in predicting possible post-operative kidney function impairment (expert opinion, EO). Moreover, among personal habits, EAU guidelines state that patients with a history of alcohol consumption > 15 g/day have a significantly higher risk of UTUC when compared with never-drinkers [22].

However, hereditary factors like Lynch and Lynch-like syndromes are equally mentioned as risk factors in all the analyzed guidelines [23,24]. EAU guidelines propose the Amsterdam criteria to identify families with a high probability of having Lynch syndrome (LE: 2A, SE: weak); the AUA panel proposes Amsterdam II criteria and refers to revised Bethesda guidelines, whilst the NCCN guidelines recommend obtaining a thorough family history in each patient with high-risk UTUC [25].

### 3.4. Diagnosis

Among all the three guidelines dealing with UTUC, only the ones by EAU clearly describe suspicious symptoms of this tumor by reporting micro/macro-hematuria, flank pain, and systemic symptoms associated with metastatic disease (anorexia, weight loss, malaise, fatigue, fever, night sweats, or cough) [26,27,28].

However, in the diagnostic work-up of suspicious UTUC, contrast-enhanced abdomen and pelvis Computer Tomography (CT) scans are recommended for investigation, offering a pooled sensitivity of 92% (CI: 0.85–0.96) and specificity of 95% (CI: 0.88–0.98) [29,30]. In case of contraindications to contrast-medium, guidelines recommend performing a Magnetic Resonance (MR) with urography, even if less sensitive and specific for the diagnosis and staging of UTUC, followed by renal ultrasound [31,32]. An integral part of the diagnostic workup of UTUC includes a careful inspection of the bladder through cystoscopy. Only the NCCN clearly recommends a direct urine sample collection for cytologic assessment (category 2A), while the other guidelines recommend urine sampling by standard collection or directly from the renal pelvis or the ureteral lumen during endourological assessment [33].

In case of uncertainty, ureteroscopy could be detrimental to the diagnosis. In this setting, the AUA panel highlights that endoscopic procedures are intended as low-impact and typically brief interventions, whereas therapeutic endoscopic procedures with curative intent are often longer with a greater risk for surgical complications. Lastly, AUA guidelines recommend the collection of bioptic specimens of suspected lesions during endoscopic procedures, whilst the EAU panel states that biopsies should preferably be avoided due to undegrading risks (SE: strong) [34,35].

The diagnostic work-up of UTUC is completed by ruling out possible organs or node metastasis with imaging. EAU recommends CT scans of the whole body for diagnosis and staging (SE: strong) or, in case of contraindications, MR urography, whilst 18F-Fluorodeoxglucose positron emission tomography/CT is preferred to assess nodal metastasis (SE: weak) [32,36]. Only NCCN includes the chest radiograph as an alternative (category 2A). In this regard, AUA guidelines are published in the non-metastatic setting, thus avoiding any assessment of secondary localization of the malignancy (Figure 1).

### 3.5. Risk Stratification and Prognosis

Clinical findings are crucial to stratify and inform patients about disease, treatment, and peri-operative management.

Apart from NCCN, guidelines recommend pre-operatively stratifying patients into low- and high-risks. According to this classification, on the one hand, the EAU recommends risk stratification to identify patients more likely to benefit from kidney-sparing treatment and those who should undergo radical treatment (LE: 3, SE: weak) [37,38]. On the other hand, the AUA/SUO panel recognizes the need for further stratification to guide the correct treatment option between ablative treatments and systemic therapy.

The main discriminating factors for the high-risk classification are positive cytology for high-grade urothelial cancer cells, local invasion at the imaging, and nodal involvement. The EAU panel further considers the high-risk group patients with a tumor size over 2 cm and/or hydronephrosis and/or multifocality [39,40]. Differently, AUA/SUO further differentiates the two risk classes according to prognosis (favorable vs. unfavorable) in the presence of urinary obstruction, multifocality or contralateral UTUC diagnosis, and involvement of the lower urinary tract [41]. Moreover, high-grade urinary cytology, suspicious nodal involvement, and the non-papillary tumor pattern are exclusive characteristics of the high-grade unfavorable UTUC [42].

Lastly, EAU guidelines also report, among risk factors for more aggressive cancer, patients’ related characteristics, such as ethnicity, with African-American patients having worse outcomes, advancing age, tobacco consumption, delayed surgery, high comorbidity and performance indices scores, and several blood-based biomarkers even if no prognostic biomarkers are validated for clinical use (LE: 3) [43,44,45,46,47]. These latter characteristics are mentioned in the AUA/SUO, but the panel considers them to be not widely available, easily identified, or measured, thereby limiting broad applicability. However, AUA recommends identifying the risk of post-operative chronic renal failure or dialysis in patients who are candidates for radical nephroureterectomy (RNU) [48].

### 3.6. Disease Management

The three guidelines deal with disease management differently (Table 1). Indeed, NCCN considers localization firstly (e.g., renal vs. other sites) and tumor grade subsequently; the AUA/SUO panel reports recommendations according to kidney sparing management vs. surgical approach; the EAU guidelines present management as per localized low-grade, localized high-grade, and metastatic disease.

#### 3.6.1. Endoscopic and Surgical Approaches

Consensus exists among EAU and AUA to adopt nephron-sparing approaches (either endoscopic or percutaneous) as first-line therapy for low-risk tumors or high-risk masses with favorable prognosis. The ureteral resection with simultaneous lymphadenectomy is indicated only in case of masses that were not completely removable endoscopically (LE: 2A; SE: strong) [49]. The NCCN recommends endoscopic management only for selected patients or those who are not suitable for RNU due to clinical or biochemical criteria; otherwise, a surgical approach is always recommended [50]. After the endoscopic management, AUA and NCCN recommend at least one instillation of Mitomycin C (or local application in the gel format) in the upper tract, able to decrease rates of local recurrence [51,52]. Accordingly, EAU reports data on Mitomycin C gel used as a single therapeutic option to chemo-ablate low-risk UTUC, whilst AUA reports considerations regarding technical procedures for upper tract instillation and to instill adjuvant pelvicalyceal chemotherapy (conditional recommendation; Grade C) or the use of a six-week induction course of BCG as the primary treatment of in situ UTUC and as an adjuvant treatment for localized non-invasive disease (EO) [53,54]. No data support the instillation of the bladder after kidney-sparing management.

In the management of high-risk non-metastatic UTUC, the guidelines reach a consensus on recommending RNU with bladder cuff excision and lymphadenectomy as the standard therapeutical approach. The kidney, together with the ureter and the bladder cuff should be removed *en block,* with the AUA panel encouraging the securing of the urinary tract in a watertight fashion to avoid the spreading of cancer cells [55]. In this context, the AUA leaves the choice of the surgical approach to the surgeon (laparoscopic or robotic minimally invasive or open), whilst according to the EAU guidelines, the open approach is more radical and safer than the laparoscopic one due to the risk of dissemination along the trocar pathway [56,57]. Moreover, AUA and EAU guidelines agree on better long-term outcomes in patients with high-risk non-metastatic UTUC who underwent regional lymphadenectomy. Otherwise, even if this procedure is recommended by the NCCN panel, data on outcome improvements are not reported.

Lastly, evidence agrees that patients who undergo RNU have a high risk of bladder recurrence (22–47%). Thus, all the guidelines recommend a post-operative single instillation of chemotherapeutic agents [58,59]. The most adopted agent is Mitomycin C and, in selected patients or in the presence of risk of intrabdominal extravasation, Gemcitabine can be used. However, the AUA panel reports that due to the absence of a direct comparison of these two agents, the choice should be based on agent availability and workflow suitable to the clinician [60,61].

#### 3.6.2. Medical Therapy

##### Neoadjuvant Chemotherapy

All the guidelines agree on the use of neoadjuvant chemotherapy with platinum-based protocols. NCCN recommends neoadjuvant chemotherapy in selected patients with both the renal pelvis and ureteral masses and with retroperitoneal lymphadenopathy, bulky (>3 cm) high-grade tumor, sessile histology, or suspected parenchymal invasion (category 2B) [62,63,64]. EAU reports that the primary advantage of neoadjuvant platinum-based therapy is to profit from better renal function before RNU and lower the disease recurrence and mortality [65,66,67]. However, the evidence presented above is not conclusive, given the significant bias and heterogeneity of the available data. Accordingly, the AUA guidelines support the use of neoadjuvant chemotherapy in patients with local or distant invasive tumors (T2–T4) or with lymph nodal involvement (N+). In particular, patients expecting an important decrease in glomerular function that would preclude post-operative platinum-based chemotherapy could benefit the most (strong recommendation; Grade B) [62,68,69,70].

##### Adjuvant Chemotherapy

In patients with UTUC who have not received neoadjuvant therapy, adjuvant cisplatin-based chemotherapy can be discussed considering that after RNU the renal function may deteriorate limiting the adoption of this strategy. To this regard, a GFR of 45 mL/min is considered a suitable threshold for the EAU guidelines panel, whilst AUA and NCCN adopt the historically used threshold of 60 mL/min (strong recommendation; Grade A) [71,72].

##### Immunotherapy

Consensus exists between the AUA, EAU, and NCCN in supporting the use of adjuvant Nivolumab in patients with high-risk muscle-invasive UTUC who had undergone radical surgery and had a tumor cell PD-L1 expression > 1% [73]. This was particularly the case in patients receiving neoadjuvant platinum-based chemotherapy or those patients who, due to comorbidities, were non-eligible for platinum-based therapies. Nonetheless, in these patients, a network meta-analysis suggests a superior oncological benefit from adjuvant platinum-based chemotherapy over immune checkpoint inhibitors [74].

### 3.7. Staging and Classification

Due to similarities between the classification and morphology of UTUC and BC, most of the guidelines do not consider a separate staging classification for the two malignancies. Indeed, only the EAU guidelines provide a complete classification of UTUC referring to the 2022 update of the 2004/2016 WHO grading classification [75] and the 8th edition of the TNM classification of malignant tumors [76].

### 3.8. Follow-Up

Follow-up in UTUC is required to detect recurrent or new primary tumors within the urothelium and regional and distant metastases. In this setting, the EAU panel states that bladder recurrence is not considered a distant recurrence.

Among sources on follow-up strategies, heterogeneity exists providing an overall low level of concordance. Similar to the approach used to describe the diagnostic management of UTUC, the three guidelines differ in reporting the follow-up criteria. In fact, NCCN reports indications according to tumor staging, EAU based on risk class, and AUA depending on the treatment chosen (Table 2 and Table 3).

After primary treatment for localized low-risk tumors, NCCN recommends, independently from surgical or endoscopic management, serial cystoscopies at 3-month intervals for the first year and, if negative, at longer intervals without specifying timing or end of the follow-up period (category 2A). Moreover, imaging of the upper tract collecting system or ureteroscopy at 3- to 12-month intervals ± abdominal/pelvic CT or MRI with and without contrast are recommended without clear timing. Similar indications for the low-risk malignancies treated with RNU come from the EAU panel, which does not indicate the need for a cystoscopy at 6 months and recommends stopping yearly cystoscopy after 5 years [77,78]. In the case of the kidney-sparing therapeutic approach, the EAU indicates a very stringent follow-up (LE: 3) with an early second-look ureteroscopy after 6 to 8 weeks and cystoscopy and CT-urography at 3 and 6 months and then yearly for 5 years (SE: weak) [79,80]. According to AUA guidelines, in the case of kidney sparing management, the early second-look should be performed within one to three months, with cystoscopy at least every six to nine months for the first two years and then annually, and upper tract imaging should be performed at least every six to nine months for two years, then annually for 5 years (EO). Meanwhile, after RNU, patients should undergo cystoscopy and cytology within three months after surgery, then every six to nine months for the first two years, and then annually for up to 5 years. Moreover, imaging should be performed at least every six to nine months for two years and then annually for up to 5 years. In the absence of recurrence, further planning of the follow-up should be based on a shared decision between the patient and the physician (EO). The consensus among the guidelines is that further investigations for distant metastasis whose risk of occurrence is very low be avoided.

For higher-grade tumors, NCCN states that follow-up should follow the same indications with the addition of chest imaging and stronger recommendation for cytology, without clearly indicating timing (category 2A). A similar strategy is recommended by EAU guidelines for high-grade UTUC treated endoscopically, with the warning to continue the follow-up beyond 5 years (SE: weak) [81]. When RNU is preferred, the EAU recommends stringent follow-up, also to detect metachronous BC, whose probability increases over time up to 4 years; after that, the risk of recurrences decreases [82,83].

In the high-risk setting, the AUA guidelines recommend a different follow-up. When the patient has undergone kidney-sparing treatment, an early cystoscopy and upper tract endoscopy with cytology should be performed within one to three months (EO) [84]. If negative for BC, patients should undergo cystoscopy and cytology every three to six months for the first three years and then annually. Upper tract endoscopy should be repeated at least at six months and one year; while imaging should be performed every three to six months for three years and then annually for up to five years. In the case of RNU, cystoscopy and cytology should be repeated at least every three to six months for the first three years and then at least annually thereafter [85,86]. Regarding post-operative imaging follow-up, AUA sub-classify according to tumor staging as follows:<T2N0M0, scans of abdomen and pelvis within 6 months and then annually (EO).>T2N0M0, multiphasic contrast-enhanced CT urography every three to six months for two years, every six months in the third year, and annually thereafter. A chest CT is indicated every 6–12 months for the first 5 years (EO).

In the absence of recurrence, further surveillance plans should be encouraged and based on a shared decision between the patient and physician.

## 4. Discussion

UTUC is a rare malignancy belonging to the family of urothelial carcinoma and shares similarities with BC. Nevertheless, only three eminent international scientific societies (NCCN, EAU, and AUA) provide clear guidelines dealing with its epidemiology, symptoms, diagnosis, treatment, and follow-up.

Our research outlined an uneven picture of reported evidence in its structure and contents. In this regard, NCCN guidelines on UTUC are presented as a chapter of Version 3.2023 guidelines on BC. Consequently, this source lacks its own important sections such as epidemiology and risk factors (apart from a brief outline on Lynch syndrome). NCCN guidelines are focused on the management of UTUC, with brief presentations of the diagnosis and follow-up of such malignancy. For this reason, the main strength point of this resource is to guide urologists easily and straightly in the correct management of the malignancy with the limitation of not providing enough evidence on its detection and prosecution of the follow-up after the first year from the main intervention.

Differently, EAU and AUA guidelines appear to be more complete resources dealing with all the aspects of UTUC, from suggestive symptoms to management and follow-up. Both these guidelines are mostly supported by strong recommendations and high-quality LE in their common points. However, our analysis highlights some differences between these eminent sources, especially on risk stratification and consequent strategies to treat UTUC. Moreover, EAU guidelines are the only ones to elaborate and report a proper TNM staging system, separated from the one of BC.

The most important difference among all the guidelines is related to follow-up timing and methods. All report different follow-up approaches according to the adopted surgical approach, with further differentiation in accordance with cancer grade. Indeed, the timing of early assessment and of performance of procedures such as cystoscopy, ureteroscopy, and imaging differs from one source to another without reaching a complete consensus. In this setting, the AUA panel released the most detailed plans of follow-up by also stratifying this based on pathological histology. Nonetheless, this detailed follow-up is composed of EOs, which were achieved by consensus of the panel, which is based on members’ clinical training, experience, knowledge, and judgment for which there is no evidence and not based on evidence and recommendations.

## 5. Conclusions

In conclusion, due to its relative rarity and similarities among urothelial carcinomas, most sources treat UTUC as a variant of BC. Indeed, among all the eminent sources created to guide physicians, only AUA, EAU, and NCCN guidelines discuss UTUC. Between these three, our research depicted high variability in the addressed sections and approaches in reporting recommendations and opinions. In this regard, we encourage urologists to always consider UTUC as an entity per se with its own diagnosis, management, and follow-up strategies. In accordance, further high-quality research is needed to gain evidence creating higher grade consensus between guidelines.

## Figures and Tables

**Figure 1 cancers-16-01115-f001:**
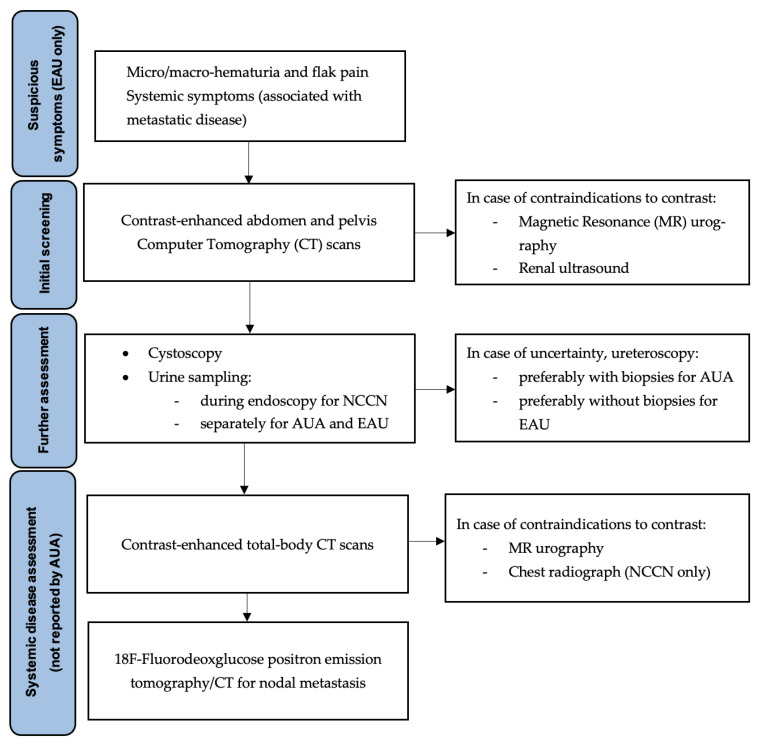
Diagnostic work-up of UTUC according to the AUA, EAU, and NCCN guidelines. Abbreviations: UTUC: upper tract urothelial carcinoma; AUA: American Urology Association; EAU: European Association of Urology; NCCN: National Comprehensive Cancer Network.

**Table 1 cancers-16-01115-t001:** Differences in UTUC management according to AUA, EAU, and NCCN guidelines.

Management Type	AUA	EAU	NCCN
Neoadjuvant platinum-based chemotherapy	Patients with local or distant invasive tumors or involved lymph nodes When GFR is expected to lower after surgical treatment	When GFR is expected to lower after surgical treatment	Patients with retroperitoneal lymphadenopathy, bulky (>3 cm) high-grade tumor, sessile histology, or suspected parenchymal invasion
Endoscopic management	Low-risk and high-risk favorable UTUC	Low-risk masses	Only for selected patients or those who are not suitable for RNU
Post-endoscopic instillation	Mitomycin C gel or BCG	Mitomycin C gel	Mitomycin C gel
RNU with bladder cuff and LND	Open/laparoscopic according to surgeon preference Remove *en block*, watertight fashion LND in high-risk UTUC	Open approach is superior for fewer risks of cells dissemination LND in high-risk UTUC	No specific indication
Post-RNU bladder instillation (Mitomycin C or Gemcitabine)	According to availability	Gemcitabine preferred in case of risk of extravasation	No specific indication
Adjuvant chemotherapy	Up to a GFR of 60 mL/min	Up to a GFR of 45 mL/min	Up to a GFR of 60 mL/min
Immunotherapy in high-risk muscle-invasive UTUC, undergone RNU and cell PD-L1 expression > 1%	No specific indication	No specific indication	No specific indication

Abbreviations: UTUC: upper tract urothelial cancer; AUA: American Urology Association; EAU: European Association of Urology; NCCN: National Comprehensive Cancer Network; RNU: radical nephroureterectomy; LND: lymph node dissection; GFR: Glomerular Filtration Rate.

**Table 2 cancers-16-01115-t002:** Summary of surveillance plan according to AUA, EAU, and NCCN guidelines in case of UTUC managed with kidney-sparing procedures.

LOW-RISK UTUC							
Diagnostic Procedure	Guideline	Early Second-Look	First Year	Second Year	Third Year	Fourth–Fifth Years	Further Assessment
Cystoscopy and cytology	AUA	Within 4–8 weeks	At 6 to 9 months	At 6 to 9 months	Yearly	Yearly	Decision made between the patient and clinician
	EAU	Within 6–8 weeks	3 and 9 months	Yearly	Yearly	Yearly	Stop after 5 years
	NCCN	NA	Every 3-months	Not specified	Not specified	Not specified	Not specified
Ureteroscopy	AUA	Within 4–8 weeks	At 6 to 9 months	At 6 to 9 months	Yearly	Yearly	Decision made between the patient and clinician
	EAU	Within 6–8 weeks	3 and 6 months	Yearly	Yearly	Yearly	Stop after 5 years
	NCCN	NA	3 to 12 months	Not specified	Not specified	Not specified	Not specified
Imaging	AUA	NA	At 6 to 9 months	At 6 to 9 months	Yearly	Yearly	Stop after 5 years
	EAU	NA	CT-urography at 3 and 6 months	Yearly	Yearly	Yearly	Stop after 5 years
	NCCN	NA	Abdominal/pelvic CT or MRI at 3 to 12 months	Not specified	Not specified	Not specified	Not specified
**HIGH-RISK UTUC**							
**Diagnostic Procedure**	**Guideline**	**Early** **Second-Look**	**First Year**	**Second Year**	**Third Year**	**Fourth–Fifth Years**	**Further Assessment**
Cystoscopy and cytology	AUA	Within 4–8 weeks	Every 3 to 6 months	Every 3 to 6 months	Every 3 to 6 months	Yearly	Encourage to continue follow-up
	EAU	Within 6–8 weeks	A 3 and 9 months	Yearly	Yearly	Yearly	Continue in accordance with the patient
	NCCN	NA	Every 3-months	Not specified	Not specified	Not specified	Not specified
Ureteroscopy	AUA	Within 4–8 weeks	At 6 and 12 months	At 6 to 9 months	Yearly	Yearly	Encourage to continue follow-up
	EAU	Within 6–8 weeks	3 and 6 months	Yearly	Yearly	Yearly	Continue in accordance with the patient
	NCCN	NA	Not specified	Not specified	Not specified	Not specified	Not specified
Imaging	AUA	NA	Every 3 to 6 months	Every 3 to 6 months	Every 3 to 6 months	Yearly	Encourage to continue follow-up
	EAU	NA	CT-urography at 3 and 6 months	Yearly	Yearly	Yearly	Continue in accordance with the patient
	NCCN	NA	Not specified	Not specified	Not specified	Not specified	Not specified

Abbreviations: AUA: American Urology Association; EAU: European Association of Urology; NCCN: National Comprehensive Cancer Network; UTUC: upper tract urothelial carcinoma; CT: Computer Tomography; MRI: Magnetic Resonance Imaging; NA: not available.

**Table 3 cancers-16-01115-t003:** Summary of surveillance plan according to AUA, EAU, and NCCN guidelines in case of UTUC managed with radical nephroureterectomy.

LOW-RISK UTUC						
Diagnostic Procedure	Guideline	First Year	Second Year	Third Year	Fourth–Fifth Years	Further Assessment
Cystoscopy and cytology	AUA	At 3 months and then at 6 to 9 months	At 6 to 9 months	Yearly	Yearly	Decision made between the patient and clinician
	EAU	3 and 9 months	Yearly	Yearly	Yearly	Stop after 5 years
	NCCN	Every 3-months	Not specified	Not specified	Not specified	Not specified
Imaging	AUA	At 6 to 9 months	At 6 to 9 months	Yearly	Yearly	Decision made between the patient and clinician
	EAU	CT-urography at 3 and 6 months	Yearly	Yearly	Yearly	Stop after 5 years
	NCCN	Abdominal/pelvic CT or MRI at 3 to 12 months	Not specified	Not specified	Not specified	Not specified
**HIGH-RISK UTUC**						
**Diagnostic Procedure**	**Guideline**	**First Year**	**Second Year**	**Third Year**	**Fourth–Fifth Years**	**Further assessment**
Cystoscopy and cytology	AUA	At 3 to 6 months	At 3 to 6 months	At 3 to 6 months	Yearly	Encourage to continue follow-up
	EAU	3 and 9 months	Yearly	Yearly	Yearly	Stop after 5 years
	NCCN	Every 3-months	Not specified	Not specified	Not specified	Not specified
Imaging	AUA (<T2N0M0)	Abdominal/pelvic CT or MRI within 6 months	Yearly	Yearly	Yearly	Encourage to continue follow-up
	AUA (>T2N0M0)	CT-urography every 3 to 6 months	CT-urography every 3 to 6 months	CT-urography every 6 months		
	EAU	CT-urography at 3 and 6 months	Yearly	Yearly	Yearly	Stop after 5 years
	NCCN	Abdominal/pelvic CT or MRI at 3 to 12 months	Not specified	Not specified	Not specified	Not specified

Abbreviations: AUA: American Urology Association; EAU: European Association of Urology; NCCN: National Comprehensive Cancer Network; UTUC: upper tract urothelial carcinoma; CT: Computer Tomography; MRI: Magnetic Resonance Imaging; NA: not available.

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
