# Peer review of "Upper Tract Urothelial Cancer: Guideline of Guidelines"

_cancers, 2024, doi:10.3390/cancers16061115_

Round 1
Reviewer 1 Report
Comments and Suggestions for Authors
The authors compare NCCN, AUA, and EAU guidelines for UTUC. Although the various guidelines were updated in 2023, this article summarizes the differences between the various guidelines very well and may be worth reading for urologists. Could you please consider only the following one point?
1. There currently seems to be no consensus on the significance of lymph node dissection during RNU. What is the current position on the treatment of lymph node dissection in each of the guidelines? If there are differences, please summarize them in Table 1.
Author Response
Dear Reviewer,
Thank you for your insightful comments. We have carefully considered your suggestion and have updated our manuscript accordingly. We've added a detailed analysis of the NCCN, AUA, and EAU guidelines' latest positions on lymph node dissection, highlighting any discrepancies among them. This update includes a new table (Table 1) that succinctly compares these guidelines, as per your recommendation. We believe these revisions significantly enhance the manuscript, providing clear guidance for urologists on this matter.
Reviewer 2 Report
Comments and Suggestions for Authors
The paper makes a significant contribution to the understanding of UTUC management by systematically comparing international guidelines. The highlighted differences among guidelines underscore the complexity of managing UTUC and the importance of recognizing it as a distinct entity with its own diagnostic and therapeutic considerations. The call for further research to improve the quality of evidence and promote international guideline harmonization is a pertinent and forward-looking conclusion.
Author Response
Dear Reviewer,
Thank you for your positive feedback on our paper. We appreciate your recognition of its contribution to the UTUC management field and agree with the need for further research to enhance guideline harmonization.
Reviewer 3 Report
Comments and Suggestions for Authors
Dear Author
Thank you for your manuscript submission. The manuscript is well-designed and well-presented.
1. What is the main question addressed by the research? The authors tried to systematically compare current international guidelines and recommendations and to identify both discrepancies andsimilarities regarding the management of UTUC. 2. What parts do you consider original or relevant for the field? What specific gap in the field does the paper address? I believe that, the authors have presented an effective and well-designed work. 3. What does it add to the subject area compared with other published material? Very professional, effective and hierarchically arrangement in the manuscript 4. What specific improvements should the authors consider regarding the methodology? What further controls should be considered? None 5. Please describe how the conclusions are or are not consistent with the evidence and arguments presented. Please also indicate if all main questions posed were addressed and by which specific experiments. The Conclusion section is very suitable and effective. 6. Are the references appropriate? Yes 7. Please include any additional comments on the tables and figures and
quality of the data.
None
Author Response
Thank you for your thoughtful review and positive feedback. We're grateful for your acknowledgment of our work's design and significance in UTUC management. Your comments bolster our confidence in the manuscript's value to the field.